# An Approach to Evaluate Pb Tolerance and Its Removal Mechanisms by *Pleurotus opuntiae*

**DOI:** 10.3390/jof9040405

**Published:** 2023-03-24

**Authors:** Priyanka Yadav, Vartika Mishra, Tejmani Kumar, Awadhesh Kumar Rai, Ayush Gaur, Mohan Prasad Singh

**Affiliations:** 1Centre of Biotechnology, University of Allahabad, Prayagraj 211002, India; priyankayadav@allduniv.ac.in (P.Y.); vartikamishra@allduniv.ac.in (V.M.); aayushgaur21@gmail.com (A.G.); 2Department of Physics, University of Allahabad, Prayagraj 211002, India; tejmani38@gmail.com (T.K.); awadheshkrai@gmail.com (A.K.R.)

**Keywords:** environmental problems, mycoremediation, *P. opuntiae*, tolerance index, biosorption, bioaccumulation

## Abstract

Widespread lead (Pb) contamination prompts various environmental problems and accounts for about 1% of the global disease burden. Thus, it has necessitated the demand for eco-friendly clean-up approaches. Fungi provide a novel and highly promising approach for the remediation of Pb-containing wastewater. The current study examined the mycoremediation capability of a white rot fungus, *P. opuntiae*, that showed effective tolerance to increasing concentrations of Pb up to 200 mg L^−1^, evidenced by the Tolerance Index (TI) of 0.76. In an aqueous medium, the highest removal rate (99.08%) was recorded at 200 mg L^−1^ whereas intracellular bioaccumulation also contributed to the uptake of Pb in significant amounts with a maximum of 24.59 mg g^−1^. SEM was performed to characterize the mycelium, suggesting changes in the surface morphology after exposure to high Pb concentrations. LIBS indicated a gradual change in the intensity of some elements after exposure to Pb stress. FTIR spectra displayed many functional groups including amides, sulfhydryl, carboxyl, and hydroxyl groups on the cell walls that led to binding sites for Pb and indicated the involvement of these groups in biosorption. XRD analysis unveiled a mechanism of biotransformation by forming a mineral complex as PbS from Pb ion. Further, Pb fostered the level of proline and MDA at a maximum relative to the control, and their concentration reached 1.07 µmol g^−1^ and 8.77 nmol g^−1^, respectively. High Pb concentration results in oxidative damage by increasing the production of ROS. Therefore, the antioxidant enzyme system provides a central role in the elimination of active oxygen. The enzymes, namely SOD, POD, CAT, and GSH, served as most responsive to clear away ROS and lower the stress. The results of this study suggested that the presence of Pb caused no visible adverse symptoms in *P. opuntiae*. Moreover, biosorption and bioaccumulation are two essential approaches involved in Pb removal by *P. opuntiae* and are established as worthwhile agents for the remediation of Pb from the environment.

## 1. Introduction

Environmental pollution caused by toxic metals (such as lead (Pb)) is out of hand because of urbanization, industrial development, agricultural practices, and other anthropogenic activities. Toxic heavy metals are the foremost environmental contaminants because of their immutable nature [1]. Pb pollution is one of the leading concerns to the environment which is mainly present in batteries, coal-burning, ore smelting, mining, industrial effluents, chemical pesticides, and gasoline additives [2]. It has the propensity to accumulate in the living system and lead to biomagnification that upraises the metal concentration in the body to such an extent that it generates toxic effects, such as osteomalacia and congenital defects, and is also hazardous to the hepatic system, renal system, and gastrointestinal and nervous systems [3]. As a consequence, the elimination of toxic metal has become an issue that needs immediate attention. Earlier, several conventional approaches including electrochemical treatment, chemical precipitation, membrane technology, and ion exchange methods have been employed for heavy metal elimination, but these practices are either expensive or effective only when applied to treat wastewater containing modest-to-high concentrations of heavy metals; thus, they are ineffective for the removal of toxic metals occurring in lower concentrations (10–100 ppm) [4]. Moreover, they may also generate toxic sludge or other secondary pollutants during the repair process that also need proper disposal [5]. In this situation, bioremediation has always been regarded a propitious approach because of both its moderation and safety attributes [6]. 

Bioremediation is an ecologically sound technology that restricts the strain on natural resources as linked to conventional methods [7]. It embraces the action of living organisms such as plants, algae, bacteria, and fungi [8]. These organisms tend to absorb toxic pollutants and show great tolerance potential under stress environments owing to their innate adsorption capability [9]. Further, they can oxidize many inorganic elements via oxidative phosphorylation that assists the removal process [10]. In addition, they display effective removal by utilizing different strategies, either via bioaccumulation or biosorption. Several microbes boast the ability to bioremediate lead while some bacteria reduce Pb stress via chelating and converting the oxidation state of Pb^4+^ into Pb^2+^ [11]. However, the overall bacterial metabolism is impacted at high temperatures because of the inactivation of numerous proteins and enzymes. Thus, to achieve the greatest amount of lead bioremediation using bacteria, it is important to take into account the suitability of bacteria to grow optimally at a given temperature while designing the onsite bioremediation [12].

Fungi including mushrooms are biota that attribute their high ability to remove metal to their versatility in unfavorable environments, great tolerance, ability to bind metal within the cell wall, and capability to produce biomass in huge quantities. Mushroom-based remediation incorporates the removal of many pollutants such as industrial wastes, pharmaceutical wastes, petroleum solid wastes, polychlorinated biphenyls (PCBs), sulfonamides, chlorinated pesticides, organic micro-pollutants, and heavy metals [13,14]. Additionally, agro-industrial wastes are used by mushrooms as substrates for growth and cultivation. Moreover, various fungal species of *Penicillium*, *Rhizopus*, *Trichoderma*, *Aspergillus*, *Agaricus*, and *Pleurotus* have manifested an association between toxic metal resistances and the capacity to reduce metal contamination from the wastewater [15,16]. For instance, *Penicillium simplicissimum* was found to tolerate 2000 mg L^−1^ Pb and 1000 mg L^−1^ Cu and it also showed removal efficiency of 16.18 and 38.98 mg g^−1^ from the liquid medium, respectively [15], whereas, *Trichoderma asperellum* displayed tolerance towards Cd, Cr, Cu, Pb, and Zn via the mechanism of biosorption [17]. *Penicillium canescens* has also been reported to accumulate As, Hg, Cd, and Pb, better when present individually [7]. Apart from this, *Aspergillus lentulus* has appeared to uptake multiple heavy metals such as Pb, Cr, Cu, and Ni concurrently [18]. Furthermore, many filamentous fungi have also been explored to accumulate heavy metals from aqueous solutions because of their high content of cell wall materials; these contents aid in improving metal binding properties [19]. *Pleurotus ostreatus* ISS-1 is a type of filamentous fungus that displays great flexibility to the stressed ecosystem [20,21]. *Pleurotus ostreatus* has been explored to possess high Pb tolerance potential and a removal capacity up to 53.7% from the liquid medium. Further, a similar study was also carried out in which *Pleurotus ostreatus* was found to reduce and remove Cr(VI) from wastewater [22,23,24]. *Pleurotus ostreatus* has also exhibited high tolerance towards Pb, Cd, and Cr, but the removal efficiency was (99.9–100.0%), (45.9–61.1%), and (29.4–64.5%), respectively. These indicated that the tolerance potential and uptake of metals by fungi varies and depends upon types of different metals [25]. Although many studies have approached removing toxic metals from wastewater via fungi, the corresponding mechanism has not been well explored yet. The present study aimed to explore a novel fungal strain, *Pleurotus opuntiae*, and investigated the tolerance behavior for Pb in the context of a tolerance index. Thereafter, different spectroscopy analyses were performed for the assessment of the accumulation of Pb by this strain from the aqueous solution and its antioxidant enzyme activity was also evaluated.

## 2. Materials and Methods

### 2.1. Establishment of Fungal Culture

The culture of *Pleurotus opuntiae* was procured from the Directorate of Mushroom Research (DMR), Solan, India, established on malt dextrose agar (MDA) media at 25 ± 2 °C and pH 6–6.5, and sub-cultured at regular intervals.

### 2.2. Screening for Pb Tolerance Characteristics of P. opuntiae

Tolerance of *P. opuntiae* towards varying concentrations of Pb was determined via the plate assay method. To conduct plate assaying, MDA media were inoculated with fungal mycelium plugs (diameter 0.5 cm) and amended with Pb salts at different concentrations (50, 100, 150, and 200) in mg L^−1^, whereas MDA plates with no metal salts served as controls. The Pb stock solutions (2000 mg L^−1^) were prepared by dissolving Pb(CH_3_COO)_2_.3H_2_O (Sigma Aldrich, Saint Louis, MO, USA) into distilled water. Thereafter, the plates were incubated at 25 ± 2 °C for 7 days to observe the growth of fungi and photographed on alternate days throughout the incubation period. The screening was performed in triplicates for each concentration. The growth of culture was calculated by image processing software (OriginPro 9.8 software (Version 2021)). Thereafter, the tolerance index (TI) was determined on the basis of the equation given below [26]:Tolerance index (TI) = Growth of fungi (mm) on metal exposure/Growth of fungi (mm) in the absence of metal exposure, at the same time interval.(1)

According to Valix et al. [27], if the TI value reaches 0 then it evidences absolute inhibition of growth towards metals; TI < 1 expresses high tolerance; TI values of 1 exhibit similar growth as the control; and TI > 1 displays absolute fungal growth in the presence of heavy metal exceeding that of the control.

### 2.3. Experimental Design for Growth, Removal, and Bioaccumulation Studies

For metal removal mechanisms, *P. opuntiae* was cultivated by following the procedure of Li et al. [28]. Briefly, the culture was allowed to grow for up to 7 d on solid MDA media at 25 ± 2 °C and, subsequently, an agar plug (5 mm in diameter) of the culture was slashed from the edge and inoculated into 250 mL Erlenmeyer flasks containing 100 mL of Pb-amended liquid MDA medium at pH 6.5. The concentrations ranged from 50, 100, 150, to 200 mg L^−1^. In the control, Pb was not mixed to the liquid medium. The fungi were incubated at 25 ± 2 °C and 180 rpm, and all the treatments were conducted at least in triplicate. After 28 d of incubation, the mycelia were collected via filtration method using Whatman filter paper no.1 and the dry biomass was measured after oven drying (60 °C and 24 h). For heavy metal analysis, mycelium and the culture medium were taken out from each Pb concentration group and the control group (media without Pb). Mycelium was also collected from each group and stored at −80 °C to analyze other parameters.

To determine the removal rate of Pb, the culture media were digested with a mixture of HNO_3_ and H_2_SO_4_ (3:1 ratio) to perform atomic absorption spectrophotometry (AAS) using standard protocols [29]. To quantify the bioaccumulation activity, the mycelium was rinsed thrice with deionized water, homogenized via a glass homogenizer, and the sample was digested with a HNO_3_ and H_2_SO_4_ mixture in a 3:1 ratio on a hot plate at 130 °C to examine the metal concentration by AAS (AAnalyst 700, PerkinElmer, Waltham, MA, USA). The percentage of Pb removed from the medium (R%) and the bioaccumulative potential (B) of fungi were calculated by using the following Equations (2) and (3), respectively.

(2)
$$R=\left(\mathrm{Ci}-\mathrm{Cf}\right)100/\mathrm{Ci}$$


(3)
$$B=\left(\mathrm{Ci}-\mathrm{Cf}\right)V/M$$

where Ci and Cf stand for initial and final Pb concentrations (mg L^−1^); V: volume of solution (L); M: dry weight of biomass (g).

### 2.4. LIBS Analysis

The qualitative analysis of elemental constituents in fungal mycelium was evaluated by laser-induced breakdown spectroscopy (LIBS). The complete arrangement of the spectroscopic technique is shown in Figure 1. The Nd: YAG (Continuum Surelite III-10, San Jose, CA, USA) laser with a repetition rate of 10 Hz, pulse duration of 4 ns, and maximum laser energy of 425 mJ at 532 nm was focused on the sample through a 30 cm focal length lens. The light radiated from the plasma was received through a small lens on top of the optical fiber and fed into the entrance slit of a grating spectrometer fitted with a charge-coupled device (CCD). The small lens on the tip of fiber optics was positioned at a 1 cm distance from the plasma. To avoid crater formation due to lasers shooting at the same location, we put the sample on the transnational stage, which moves in all three directions (X, Y, Z), to obtain a fresh surface that helps to maintain the same lens distance to the sample, which is an important parameter for LIBS.

### 2.5. SEM, FTIR, and XRD Analysis

Scanning electron microscopy (SEM) analysis was conducted to examine the effect of heavy metals on the surface morphology of fungal mycelium [30]. The colonies of fungal mycelium were excised from the periphery and fixed in 3% glutaraldehyde (in 0.1 M phosphate buffer, pH 7.4) and incubated at 4 ± 2 °C for 12 h. The samples were then rinsed with 0.1 M phosphate buffer (pH 7.4) three times for 5 min, following the dehydration series in ethanol solutions at 70%, 80%, 90%, and thrice in 100% for 15 min each. In a desiccator, the samples were dried at 25 ± 2 °C, coated with carbon, and subjected to Jeol-EPMA scanning electron microscopic analysis (JXA-8100, Tokyo, Japan).

The form of functional groups required in Pb uptake on the surface of *P. opuntiae* was characterized by Fourier Transform Infra-Red (FTIR) according to the procedure explained by Gola et al. [31]. Briefly, the fungal mycelia were harvested at the end of experiments and freeze dried at −85 °C under high vacuum conditions using a freeze dryer; then, 1 mg fine powder of lyophilized fungal mycelia was evenly mixed with 100 mg KBr powder. All the studies were performed using a PerkinElmer (USA) FT-IR/FIR frontier spectrometer in the range of 500–3500 cm^−1^. 

The X-ray diffraction (XRD) technique was performed to establish the crystalline structure of Pb in the fungal biomass. After the end of treatments, mycelium was collected and freeze dried. Then, the dried samples were grinded to powder to perform XRD analysis (Rigaku, Tokyo, Japan, SmartLab 3 kW).

### 2.6. Analysis of Proline Content and Malonaldehyde (MDA)

The effect of Pb on proline content was evaluated at different concentrations of Pb in *P. opuntiae*, according to the procedure explained by Gratao et al. [32].

The total MDA concentration was measured following the procedure reported by Li [33]. The homogenate of fungal mycelium (0.2 g) was prepared in 5% TCA (1.5 mL) and, thereafter, centrifuged at 6000 rpm for 20 min. An aliquot of supernatant (0.5 mL) was mixed in 20% TCA (1 mL) and 0.5% thiobarbituric acid (TBA) (1 mL), and then heated at 90 °C for 30 min. Afterwards, the mixture was cooled and centrifuged again at 12,000× *g* for 15 min. The absorbance was determined at 450 nm, 532 nm, and 600 nm, respectively, and MDA concentration was calculated as follows:
(4)
$$\mathrm{MDA}=6.45\left(A532-A600\right)-0.54\;A450$$


### 2.7. Analysis of Enzymatic Antioxidants in Pb Containing Mycelia

To analyze the antioxidant enzymes, the fungal mycelium was homogenized in ice-cold 0.1 M phosphate-buffered saline (7.0 pH) utilizing a Potter–Elvehjem glass homogenizer equipped with a Teflon pestle. The homogenized mycelia were centrifuged firstly at 2500 rpm for 10 min, at 4 °C, and the supernatant was collected followed by a second centrifuge at 12,000× *g* for 20 min at 4 °C. Then, these supernatants were stored at −80 °C for further analysis. The total proteins were estimated by following the procedure of Lowry et al. [34]. The SOD activity in the mycelium was evaluated by following the method reported by Das et al. [35]. The unit of SOD activity was indicated in U mg^−1^ of protein.

To determine catalase activity, the Aebi [36] method was performed and also demonstrated the capability of the enzyme to degrade H_2_O_2_ to form H_2_O and molecular oxygen. The CAT assay was expressed in U mg^−1^ of protein.

A reduced glutathione assay was performed according to the procedure explained by Moron et al. [37] and the POD assay was conducted according to the method reported by Zhu et al. [38]. POD was expressed in U mg^−1^ of protein.

### 2.8. Statistical Analysis

To enhance the analytical precisions of the variables, all the experiments were performed in triplicates and the obtained data were statistically analyzed using OriginPro 9.8 software (Version 2021) with One-Way Analysis of Variance (ANOVA) and the comparison of means was carried out using Tukey test (*p* ≤ 0.05).

## 3. Results and Discussion

### 3.1. Establishing Tolerance of P. opuntiae to Pb

The tolerance index (TI) of the macrofungi in the presence of varying concentrations of Pb is represented in Figure 2. The results showed that *P. opuntiae* fully adapted to Pb stress and could effectively tolerate up to 200 mg L^−1^ Pb. The TI values remained moderate to high when exposed to varying time periods and concentrations. The total biomass of *P. opuntiae* grown in Pb did not have much difference as compared with the control biomass. These results are most probably owing to the stimulatory effect of Pb on fungi, which was also reported by Anahid et al. [39]. In the present study, the maximum tolerance (TI = 0.95) was observed at 50 mg L^−1^ Pb concentration after 48 h of incubation and, thereafter, the TI values declined slightly when reaching the climax of the rapid phase with an increasing concentration (50 mg L^−1^ to 200 mg L^−1^) of Pb; this might be owing to metal in low concentrations being used as a nutrient, which, in return, does not instigate further stress, while the absorption of heavy metal also aided in the evolution of some tolerance mechanisms. A similar study was also discussed by Wang et al. [40] in which the biomass of *Pleurotus ostreatus* ISS-1 was found to be maximum compared with that of the control on the ninth day. This could be due to the Pb stress that triggers oxalic acid secretion by fungi and precipitates lead oxalate to lessen the toxicity and enhance the growth of fungi [41]. *Pleurotus ostreatus* HAAS revealed the tolerance to Pb, but concentrations above 500 ppm led to toxic effects and resulted in less fungal growth with deformed hyphae in Pb-incorporated media [25]. Thus, further increase in the Pb concentrations could hinder the growth of macrofungi due to the accession in the duration of the lag phase and many biological factors [42]. Additionally, prolonged exposure to higher Pb concentrations might result in damage to cells through thiol binding, denaturation of proteins, and the replacement of essential metals associated with biological reactions [43]. 

### 3.2. Removal and Bioaccumulation Efficiency of Pb by P. opuntiae

The Pb removal potential of *P. opuntiae* was performed in aqueous conditions and exposed to different concentrations ranging from 50 to 200 mg L^−1^. In this study, the result indicated that the removal rate progressively increased until it achieved 99%. Figure 3 shows that the highest removal rate of 99.08% was recorded for Pb at 200 mg L^−1^, whereas a 74.86% removal rate was observed at 50 mg L^−1^ concentration. This suggests that with an increase in the Pb concentration, the removal rate also increased significantly up to 200 mg L^−1^. Yang et al. investigated varying concentrations of Pb, Cd, and Cr, and the maximum removal rate was observed for Pb by *Pleurotus ostreatus* [25]; even though it had the highest concentration in the media, an instant decrease to 53.7% was observed at 500 mg L^−1^ of Pb in *Pleurotus ostreatus* [40]. Therefore, it could be considered that at defined higher concentration of metals, the removal rate is also decreased. This decrease in the removal percentage with increasing Pb concentrations might be because of the lesser recovery of biomass that reduces the accessibility of functional group [44]. However, a variety of fungal species have been reported with great adroitness in the removal of Pb from the aqueous systems. For example, the removal rates of Pb for *Phlebia brevispora* and *Penicillium polonicum* have been recorded as 97.5% and 90.3%, respectively, under optimal conditions [45,46]. *T. asperellum* and *S. chinense* have been reported to remove 57.1% and 80.6%, respectively, at the concentration of 100 ppm Pb from aqueous solution [30]. Many studies have reported that fungi might neutralize the toxicity of Pb via the appropriate mechanism of extracellular biosorption and intracellular bioaccumulation. Additionally, it is also suggested that several macrofungi could accumulate ample toxic heavy metals inside their cells without impairing their integrity [47,48]. In the present study, *P. opuntiae* expressed better bioaccumulation potential of Pb (Figure 3). The bioaccumulation of Pb was significantly enhanced with increasing concentrations and the highest uptake reached 24.59 mg g^−1^ at 200 mg L^−1^ Pb concentration. The results are in accordance with a prior study by Mishra and Malik [18], suggesting that the bioaccumulation ability of fungi was correlated with the concentrations of heavy metals. Several species have been explored to investigate the accumulation capability in the presence of Pb; the uptake capacity of *A. niger*, *P. simplicissimum*, and *T. asperellum* were 0.054 mg g^−1^, 0.038 mg g^−1^, and 0.017 mg g^−1^, respectively, at 250 mg L^−1^ Pb concentration [15]. However, *F. velutipes* results in a minimal increase in Pb uptake (0.707 mg g^−1^) at 4 mM concentration [49]. According to Chen et al. [30], living cells of *T. asperellum* and *S. chinense* exhibited a significant result (17.52 mg g^−1^ and 23.05 mg g^−1^, respectively) when exposed to 200 ppm of Pb concentration, whereas a reduced uptake (7.50 mg g^−1^ and 11.52 mg g^−1^, respectively) was observed at 50 ppm. A similar finding was also reported in which the maximum intracellular accumulation (14.2 mg g^−1^) was observed in *Pleurotus ostreatus* at 200 ppm [25]. The accumulation of Pb has also been documented in other strains of filamentous fungi, for instance, *Galerina vittiformis*, *A. lentulus*, and *P. chrysosporium* [11]. These findings confirmed that at low heavy metal concentrations, there might exist an ion exchange between the binding site of cells and metal ions, whereas an increased metal concentration could have aided to subdue the restriction in mass transfer between the fungal cell wall and heavy metal solution, thereafter licensing further collisions in between them to obtain a maximum uptake of metals [18]. Thus, bioaccumulation is revealed as an essential mechanism in the reduction of heavy metals from contaminated systems via the interaction of heavy metals with sulfur-containing peptides and converting them into nontoxic metallothionein [47]. However, the transportation of Pb inside the cell occurred with high affinity through the intracellular binding of metal and binding sites present on the cell surface, providing a driving force in the intracellular uptake of metal. These processes are most likely to be generated as detoxification mechanisms within the fungal cell [21].

### 3.3. LIBS Analysis

The LIBS spectra of fungal mycelium with and without Pb stress are shown in Figure 4. The wavelength of LIBS spectra ranges from 240 to 850 nm, indicating the elementary composition of *P. opuntiae*. Spectral lines were identified using the National Institute of Standards and Technology (NIST) atomic spectroscopic database [50]. Spectral lines of Ca, Fe, K, Na, Mg, Pb, C, N, and O were present in the LIBS spectra. On the basis of the NIST database, four Pb emission lines at 357.2 nm, 363.9 nm, 368.3 nm, and 405.7 nm were observed in the Pb-stressed mycelium. The emission lines of Fe and K were found as substituted by Pb. Furthermore, LIBS emission lines of Ca at 396.7 nm and 422.6 nm were also observed near the Pb emission lines. Additionally, the intensities of C, Mg, Na, and H were observed to be lower in Pb-stressed mycelia compared with the control. Furthermore, the LIBS patterns of some elements such as N and O remained unchanged in their intensity after exposure to Pb stress. Many studies have reported the elemental composition in plants by LIBS technology, such as the presence of Cr in rice leaves, Pb in cabbage, Pb in *Curcuma longa*, and Cu in tobacco leaves [51,52,53]. Nevertheless, this is the first approach that focused on analyzing the change in the elemental composition of filamentous fungus *P. opuntiae* under Pb stress.

The quantitative analysis of fungal mycelium grown under different concentrations of Pb was calculated via the atomic absorption spectroscopic (AAS) technique and its data were compared with LIBS intensities. Both the results followed a similar trend, that is, increased Pb accumulation was observed in mycelium with increasing Pb concentrations. Hence, in Figure 5, it can be concluded that both the results are in good agreement.

**Figure 4 jof-09-00405-f004:**
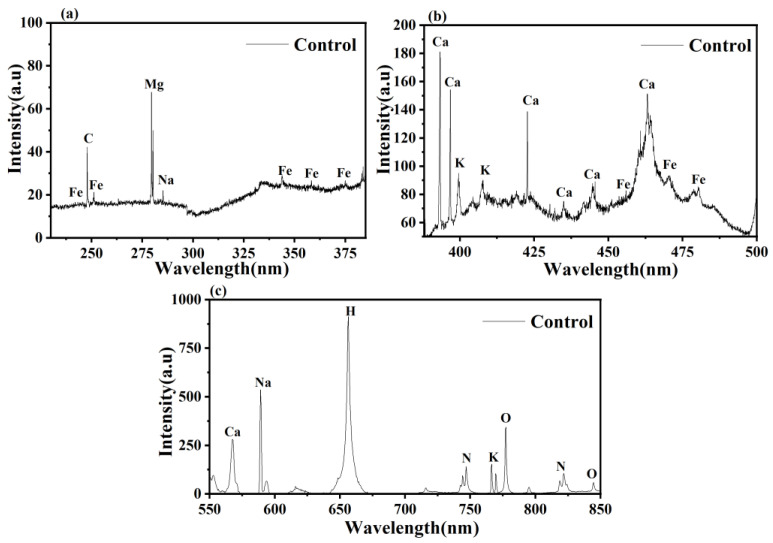
LIBS spectra of *P. opuntiae*, ranging from 250 nm to 850 nm wavelength, grown in the absence of Pb (**a**–**c**) and grown in the presence of Pb at 200 mg L^−1^ concentration (**d**–**f**).

### 3.4. SEM Analysis

In the present study, changes in the surface morphology of *P. opuntiae* mycelium grown in the absence and under the exposure of Pb were examined. Figure 6a–c depicts the image of control mycelium whereas Figure 6d–f) illustrates the image of *P. opuntiae* mycelium after exposure to Pb at 200 mg L^−1^ concentration. In the control micrographs, enlarged, string-like hyphae appeared with intertwined branches, consisting of flocculent and porous structures and thus forming a meshwork in an irregular manner. However, Pb-exposed mycelium displayed distorted hyphae that were undistinguishable, compactly packed without pores, and resulted in an even surface which might be owing to the adsorption of Pb onto the surface of mycelium. Liu et al. suggested that changes in hyphae morphology and compaction after metal exposure could result from a detoxification mechanism [54]. Many studies have also shown similar findings in which changes were observed unlike the control group [46,55]. These findings evidence that the bioaccumulation of Pb could occur after adsorption on the surface of mycelium.

### 3.5. FTIR Analysis

The interaction of Pb with several biomolecules of the cell wall in *P. opuntiae* is indicative of a detoxification mechanism involving metal accumulation and biosorption processes in metal ion removal. The FTIR spectra aided the identification and existence of functional groups in the mycelium of *P. opuntiae* before and after Pb exposure. The main functional groups associated with different peaks were identified and are listed in Table 1; the involved functional groups of mycelium include hydroxyl, amino, nitro, carbonyl, phosphoryl, and other groups [56]. In the present study, the FTIR spectra of control mycelium are represented in Figure 7a in which the band between 3500 cm^−1^–3000 cm^−1^ represents a strong absorption band of N–H and O–H stretching. The peak at 2356 cm^−1^ is representative of C–O stretching of the carbonyl compound, 1737 cm^−1^ is assigned to C=O group stretching in ester, 1374 cm^−1^ appeared as an amide III group, 1217 cm^−1^ denote the phosphate group, and the peak at 1024 cm^−1^ belongs to the C–O–P, C–C, C–O–C, and C=C groups of saccharides [55,57]. However, in the presence of Pb stress (Figure 7b), a significant shift was observed in the absorption peak. All the functional groups were similar in both spectra with some shifting of wavenumber. Further, two additional peaks also emerged at 527 and 673 cm^−1^ that are ascribed to nitro compounds, disulfide groups, and C=O stretching in amides, respectively, under Pb stress.

The changes in absorption bands detected by FTIR analysis in the mycelium of *P. opuntiae* confirmed the participation of the cell wall in Pb removal via bioaccumulation and biosorption mechanisms. Biosorption of metal ions via fungal biomass appeared to employ several functional groups [58]. Different functional groups being utilized in the binding of many metal ions results in the elimination of toxic metals from the ecosystem. For instance, Cu removal occurs via functional groups such as carboxyl, amino, and phosphate groups of *Mucor rouxii* [15], and Cd adsorption implies carboxyl, hydroxyl, and amino groups found in the biomass of *P. simplicissimum* [59]. Therefore, it could be suggested that the negatively charged functional groups including hydroxyl, amino, halide, carboxyl, nitro, and phosphate groups offer electrostatic forces in the binding of positively charged Pb to the cell’s surface and protect the cells from metal toxicity.

**Table 1 jof-09-00405-t001:** The FTIR frequencies (cm^−1^) and apportionment of functional groups present in *P. opuntiae* before and after treatment of 200 mg L^−1^ Pb.

Control	Pb (200) mg L^−1^	Functional Groups
3292	3289	O–H and N–H stretching
2356	2352	C–O stretch of carbonyl compound
1737	1735	C=O group of ester
1374	1364	Amide III group
1217	1216	Phosphate group
1024	1015	C–C, C=C, C–O–C, C–O–P of saccharides
-	673	C=O in amides
-	527	Nitro compounds and disulfide groups

**Figure 7 jof-09-00405-f007:**
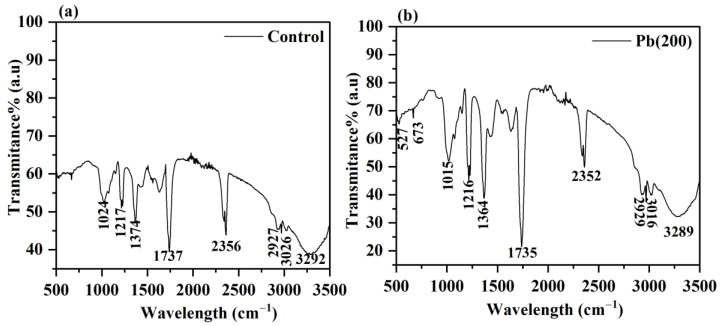
FTIR spectra of *P. opuntaie* grown in control (**a**) and under the exposure of Pb at 200 mg L^−1^ concentration (**b**).

### 3.6. XRD Analysis

XRD analysis provides information on the constituents of the mineral complex on the surface of mycelium and also aids in determining the form of the sample, whether it is amorphous or crystalline. In Figure 8, the XRD pattern of control indicated the amorphous structure of mycelium, whereas the monocrystalline form appeared because of the formation of a mineral complex after Pb exposure. This phenomenon also demonstrated a strong correlation between Pb and S, suggesting the transformation of Pb ion into a mineral particle as PbS and could be identified via PDF#77-0244. These might form because of the substitution of the hydrogen atom from –SH by Pb. A similar result was also found by Wang et al. [40]. However, other forms of Pb such as lead hydroxyphosphate [Pb_5_(PO_4_)_3_OH] and pyromorphite [Pb_9_(PO_4_)_6_] were also observed after the process of biomineralization through the mycelium of *Phanerochaete chrysoporium* and *Auricularia pumila*, respectively [60,61].

### 3.7. Effect of Pb Stress on Proline Content and MDA

Proline has been identified as an optimistic indicator and one of the primary metabolic responses to stress. Furthermore, it has the ability to detoxify free radicals via the formation of stable complexes, and their maximum accumulation assists organisms to reduce oxidative stress [62]. In this study, an increased proline level was observed with increasing Pb concentration (Figure 9a), suggesting that an increment in oxidative stress could result in a high accumulation of proline. This indicated that proline ensures protection to enzymes and biological membranes or the instant rise in proline content might rather aid enhanced bioaccumulation of Pb within the hyphae.

MDA has appeared as a measure of oxidative stress via indicating the degree of lipid peroxidation, extends of damage in the membrane, and is also considered a secondary product of lipid peroxidation when exposed to ROS, and thus is regarded to act as a marker of physiological stress [63]. In this study, the value of MDA was found as significantly increased in *P. opuntiae* under the stress of Pb. Except at lower concentrations of Pb, no obvious increase was observed in MDA level as compared to control. Thereafter, an elevated level of 8.77 nmol g^−1^ MDA was recorded at 200 mg L^−1^ concentration of Pb (Figure 9b). A similar study also reported that an increased concentration of MDA was observed under the exposure of higher concentrations of Cd and Cr stress in *Pleurotus ostreatus* HAU-2 [28]. Hence, these results suggested that the maximum concentration of Pb resulted in peroxidation of the biological membrane of macrofungi.

**Figure 9 jof-09-00405-f009:**
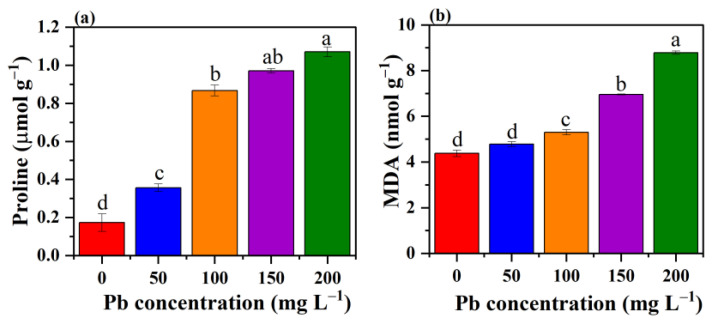
The effect of various concentrations of Pb on the proline content (**a**) and MDA level (**b**). Values followed by different letters in the same figure differ significantly (*p* < 0.05).

### 3.8. Effect of Pb Stress on the Antioxidant System of P. opuntiae

Heavy metals can adversely affect organisms at several levels by inducing many toxic responses such as damage to cellular components, depletion of enzyme activity, and denaturation of nucleic acid and proteins. With the increasing concentration of heavy metals, a significant rise in ROS production leads to oxidative stress and results in several dysfunctions and tissue damage [64]. As a result, fungi respond to oxidative stress by protectively scavenging ROS and conserving antioxidant defense compounds. SOD, POD, and CAT are essential parts of the cellular immune system that have a remarkable impact on the reduction of cellular active oxygen [65]. In the present study, the SOD activity appeared to be increased first with rising Pb concentrations (up to 100 mg L^−1^); later, the SOD activity was found to decrease at 200 mg L^−1^ concentration as compared with the control. Figure 10a showed the SOD activity recorded a maximum value of 237.8 U mg^−1^ proteins for Pb at 100 mg L^−1^ concentration. The POD and CAT activities were observed as mostly higher than the control, and a steady rise in the values of POD and CAT was also obtained with increasing concentrations of Pb. However, a maximum value for POD activity was noted (64.1 U mg^−1^ protein) for Pb at 200 mg L^−1^ (Figure 10b). Similarly, the CAT activity reached the highest value (10.5 U mg^−1^ protein) for Pb at the concentration of 200 mg L^−1^ (Figure 10c). The profile was similar to *Fusarium solani*, which resulted in enhanced values for SOD, POD, and CAT activities under the exposure of Cd and Cr [66]. The rise in SOD, CAT, and POD activities exhibited a defensive role in fungi generated because of the toxicity of Pb. These results suggested that these three enzymes are directly linked to ROS reduction [28,65].

GSH performs a key role in the antioxidant system and is involved directly in the reduction of ROS released during stress. On exposure to high metal toxicity, an elevated concentration of GSH ensures detoxification of heavy metals, keeps the cellular redox state in balance, and also participates in the regulation of H_2_O_2_ level [67,68]. In this study, GSH levels increased significantly on exposure to high Pb stress. A maximum GSH concentration 11.1 µg g^−1^ was observed at 200 mg L^−1^ Pb (Figure 10d). However, at lower Pb concentrations, the amount of GSH was quite similar compared to the control. The increasing trend of GSH levels in the presence of Cd was also reported in *Pleurotus ostreatus* [28]. A similar result was also reported for some plants such as *Sedum alfredii* and *Arabidopsis trichome*, in which elevated levels of intracellular GSH were observed because of metal exposure [69,70]. These results suggest an enlarged production of GSH leads to intracellular chelation of metal ions, enabling subcellular compartmentalization, and thus plays an important function in Pb detoxification.

**Figure 10 jof-09-00405-f010:**
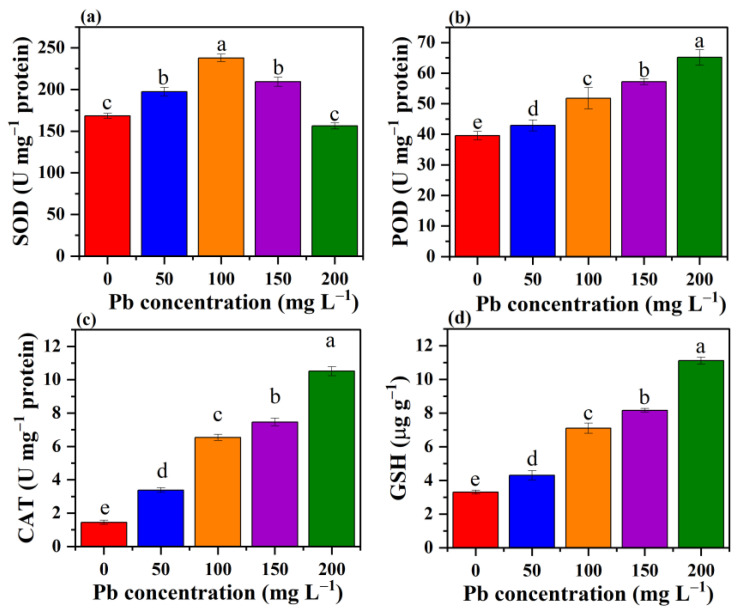
The effect of various concentrations of Pb on the activities of SOD (**a**), POD (**b**), CAT (**c**), and the amount of GSH (**d**). Values followed by different letters in the same figure differ significantly (*p* < 0.05).

## 4. Conclusions

In the present study, a filamentous fungus *P. opuntiae* showed enhanced tolerance potential toward increasing Pb stress concentrations. The strain could effectively tolerate up to 200 mg L^−1^ Pb and removed 99.08% from the aqueous medium via involving biosorption and bioaccumulation mechanisms. However, a high concentration of Pb contributed to a slight variation in the surface morphology of mycelium. The LIBS patterns indicated four Pb emission lines at 357.2 nm, 363.9 nm, 368.3 nm, and 405.7 nm. FTIR analysis revealed certain functional groups on the outer surface of cells that provide sites for Pb binding. XRD analysis resulted in the transformation of Pb ions into mineral complexes in the form of PbS. The Pb stress induced ROS generation and caused oxidative damage, but the fungal cell coped with the Pb stress by producing antioxidant enzymes, namely, SOD, POD, CAT, and GSH in higher concentrations than the control to clear away ROS and repair the oxidative damage. However, an increased amount of GSH displayed a crucial role in Pb accumulation. These findings suggested that the macrofungi *P. opuntiae* are therefore promising agents for the removal of Pb from the aqueous medium and also aid in clarifying the mechanisms involved in Pb removal.

## Figures and Tables

**Figure 1 jof-09-00405-f001:**
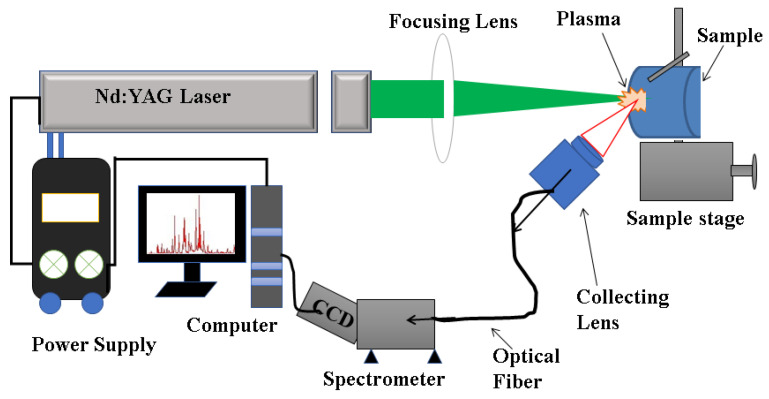
Schematic arrangement of laser-induced breakdown spectroscopy.

**Figure 2 jof-09-00405-f002:**
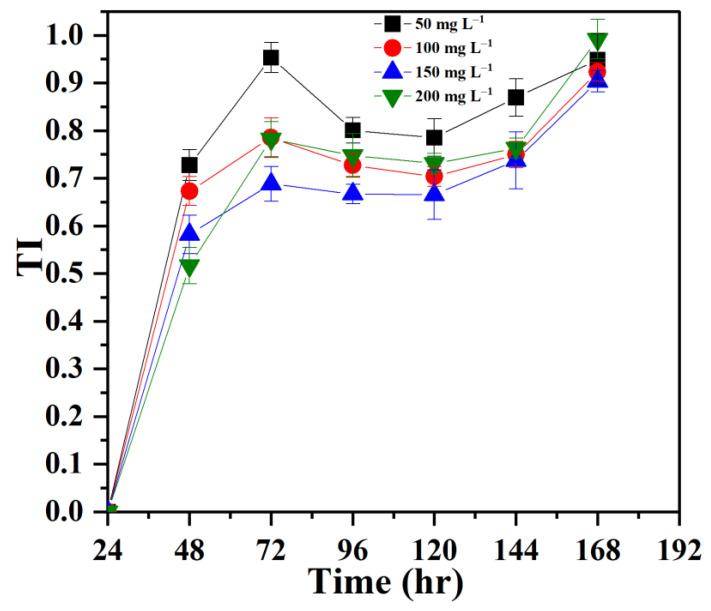
Tolerance index of *P. opuntiae* under various concentrations of Pb in MDA media after 7 days at 25 ± 2 °C.

**Figure 3 jof-09-00405-f003:**
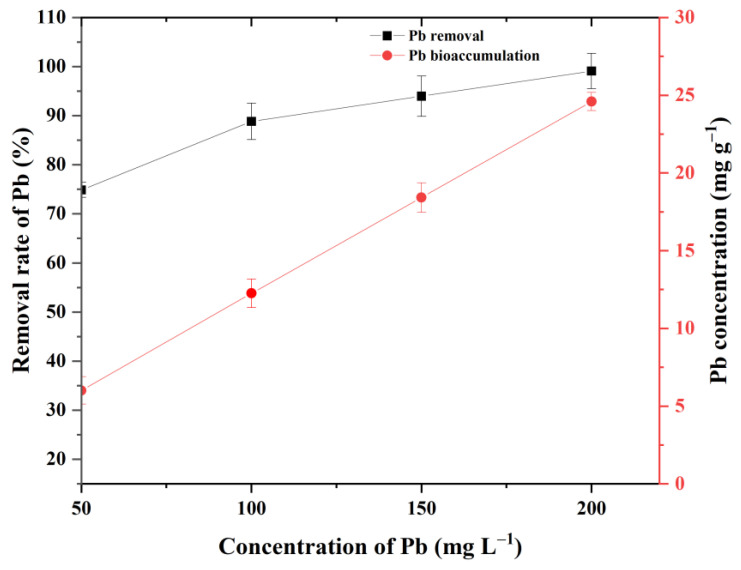
The removal rate and bioaccumulative potential of *P. opuntiae* after 28 d of incubation at 25 ± 2 °C and 180 rpm in liquid MDA media at 6.5 pH.

**Figure 5 jof-09-00405-f005:**
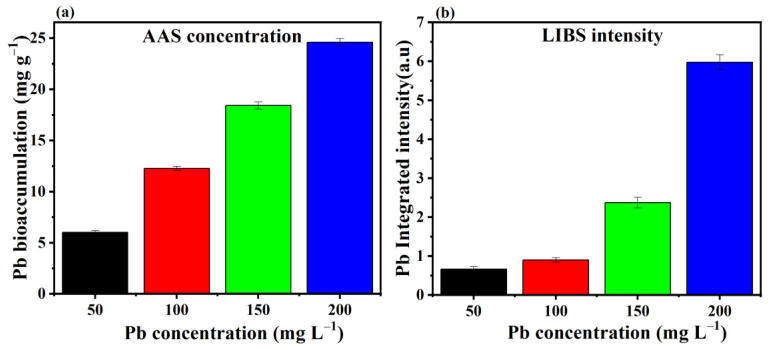
Comparison of different concentrations of Pb using atomic absorption spectroscopy (**a**) with LIBS intensity (**b**).

**Figure 6 jof-09-00405-f006:**
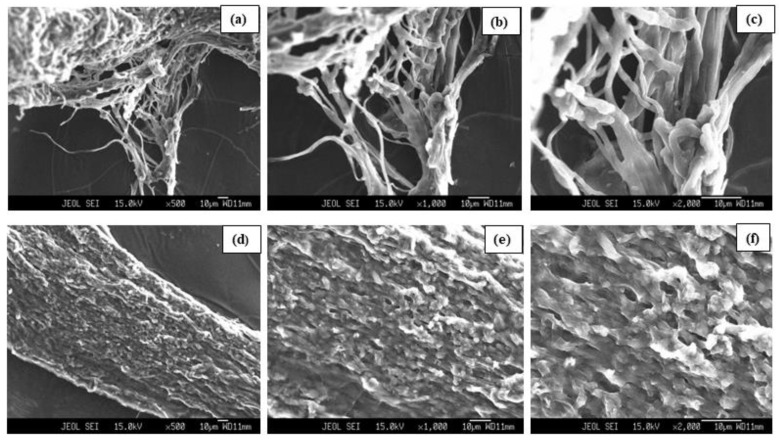
SEM images representing the surface morphology of *P. opuntiae* grown in control (**a**–**c**) and Pb at 200 mg L^−1^ concentration (**d**–**f**).

**Figure 8 jof-09-00405-f008:**
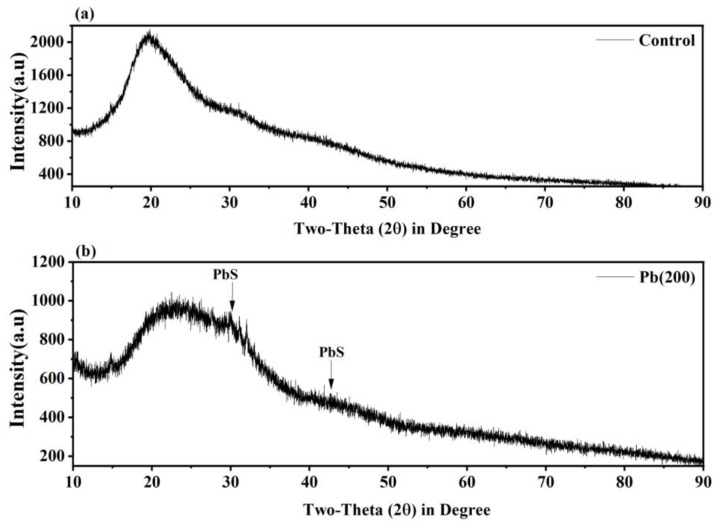
XRD spectra of *P. opuntaie* grown in control (**a**) and under the exposure of Pb at 200 mg L^−1^ concentration (**b**).

## Data Availability

The data presented in this study are available in article.

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
