# Peer review of "An Approach to Evaluate Pb Tolerance and Its Removal Mechanisms by Pleurotus opuntiae"

_jof, 2023, doi:10.3390/jof9040405_

Round 1
Reviewer 1 Report
The manuscript entitled ‘An approach to evaluate Pb tolerance and its removal mechanisms by Pleurotus opuntiae’ is very well written and collated manuscript. The author has put together the findings of the research in simplified language that is easy to apprehend.
The manuscript can be a good piece of information for the readers however, a few minors query and revisions must be addressed:
Figure 5 should show the data of control, why was it not included.
· Why did author chose two approaches to analyze metals?
· Line 53-69: avoid definitions. This section can be reduced, rather bioremediation of Pb can be discussed.
· Line 77: expression of the unit can be corrected. Across the manuscript mg/g has been used except this one mg g-1
· Pleurotus ostreatus and P. ostreatus both are used in the manuscript, please use any of them uniformly
· Line 136: digested using what?
· Line 168: correct fonts
· Line 255-56: revise the sentence
· Line 293 Spacing: P.opuntiae
· Line 379-80: correct chemical formula
Author Response
Figure 5 should show the data of control, why was it not included.
Response: The authors are grateful to the reviewer for their insightful suggestions. The authors concluded that the fungal mycelium was not grown in the presence of Pb. Therefore, in Figure 5, the control data has not been included.
Why did author chose two approaches to analyze metals?
Response: The authors are delighted to justify the reasonability of choosing two approaches.The LIBS technique allowed for the simultaneous analysis of numerous elements whereas, the AAS technique required the use of distinct standards for each element's study. Also, these approaches aid in the correlation of LIBS data with AAS data.
Line 53-69: avoid definitions. This section can be reduced, rather bioremediation of Pb can be discussed.
Response: The authors are grateful to reviewer for their valuable suggestion. We have discussed the bioremediation of Pb within the manuscript.
Line 77: expression of the unit can be corrected. Across the manuscript mg/g has been used except this one mg g-1
Response: The authors are thankful to the reviewers and the expression of the unit has been corrected as per reviewer suggestion.
Pleurotus ostreatus and P. ostreatus both are used in the manuscript, please use any of them uniformly
Response: The authors have replaced the P. ostreatus to Pleurotus ostreatus in entire manuscript.
Line 136: digested using what?
Response: The authors as per valuable suggestion of reviewer have added the complete digestion procedure in the manuscript.
Line 168: correct fonts
Response: Fonts have been corrected as per reviewer’s suggestion.
Line 255-56: revise the sentence
Response: The authors made corrections to make the article more comprehensible and easier to understand.
Line 293 Spacing: P.opuntiae
Response: The authors are grateful to the reviewer for highlighting the improvements that are necessary. The authors made the necessary corrections and rectified typing errors as per reviewers’ valuable suggestion.
Line 379-80: correct chemical formula
Response: The chemical formula has been corrected as per reviewer’s suggestion.

Reviewer 2 Report
As far as I know, it is great work!
If possible, pls explain why choose Pleurotus opuntiae as the tested strain.
Please mark the multiple comparison results in Fig 3, 5.
and to be sure that the font size of fig 9 and 10 is suitable.
Author Response
If possible, pls explain why choose Pleurotus opuntiae as the tested strain.
Response: The authors are delighted to justify the reasonability of selecting Pleurotus opuntiae as the tested strain. The authors described the reason that Pleurotus opuntiae is an unexplored species in the field of mycoremediation.
Please mark the multiple comparison results in Fig 3, 5.
Response: The authors have tried their best to incorporate the suggestions in Figure 3 and 5. At the same time authors are grateful to the reviewers for their valuable suggestions which we have incorporated as per the need.
And to be sure that the font size of fig 9 and 10 is suitable.
Response: The authors are grateful to the reviewer for highlighting the improvements that are necessary. The authors made the necessary corrections and rectified the font size of fig 9 and 10 as per reviewers’ valuable suggestion.

Reviewer 3 Report
Dear Editors and Authors,
Thank you for the invitation to review the manuscript entitled: "An approach to evaluate Pb tolerance and its removal mechanisms by Pleurotus opuntiae". In my opinion. It is an important paper suitable for the journal's aims and scope. The figures are really interesting and nice to see for the reader. I have a few minor comments/questions:
- why don't you use the term "mycoremediation" in your work? Maybe it's worth adding them at least in keywords to increase the chance that a potential reader will find your manuscript? is a fairly common term for using mushrooms to remediate pollutants;
- lines 70-97 - please start with the introduction of what kind of utilities/wastes are remedied with mushrooms, then describe wastewaters in the most detail;
- line 77 - please use superscript in the unit; use this form of units for the whole manuscript (remove "/ ");
- Figure 6 - is it possible to put a-c in the first line and then d-f in the second line - it could be clearer (control in one line and Pb in the second line);
Best regards,
Reviewer
Author Response
Why don't you use the term "mycoremediation" in your work? Maybe it's worth adding them at least in keywords to increase the chance that a potential reader will find your manuscript? is a fairly common term for using mushrooms to remediate pollutants.
Response: The authors are thankful to reviewer for their insightful suggestion. The term “mycoremediation” has been added in abstract section as well as in keywords within the manuscript.
Lines 70-97 - please start with the introduction of what kind of utilities/wastes are remedied with mushrooms, then describe wastewaters in the most detail.
Response: The authors are grateful to reviewer for their valuable suggestion. We have revised the introduction as per reviewers’ suggestion.
Line 77 - please use superscript in the unit; use this form of units for the whole manuscript (remove "/ ").
Response: The authors as per valuable suggestion of reviewer made significant changes through adding the superscript in the units within the manuscript.
Figure 6 - is it possible to put a-c in the first line and then d-f in the second line - it could be clearer (control in one line and Pb in the second line).
Response: The authors have modified the figure as per reviewer’s suggestion.
